# Efficient, Recyclable, and Heterogeneous Base Nanocatalyst for Thiazoles with a Chitosan-Capped Calcium Oxide Nanocomposite

**DOI:** 10.3390/polym14163347

**Published:** 2022-08-17

**Authors:** Khaled D. Khalil, Hoda A. Ahmed, Ali H. Bashal, Stefan Bräse, AbdElAziz A. Nayl, Sobhi M. Gomha

**Affiliations:** 1Department of Chemistry, Faculty of Science, Cairo University, Giza 12613, Egypt; 2Department of Chemistry, Faculty of Science, Taibah University, Al-Madinah Almunawarah, Yanbu 46423, Saudi Arabia; 3Institute of Organic Chemistry (IOC), Karlsruhe Institute of Technology (KIT), Fritz-Haber-Weg 6, 76133 Karlsruhe, Germany; 4Institute of Biological and Chemical Systems-Functional Molecular Systems (IBCS-FMS), Director Hermann-von-Helmholtz-Platz 1, 76344 Eggenstein-Leopoldshafen, Germany; 5Department of Chemistry, College of Science, Jouf University, Sakaka 72341, Saudi Arabia or; 6Department of Chemistry, Faculty of Science, Islamic University of Madinah, Madinah 42351, Saudi Arabia

**Keywords:** chitosan, calcium oxide nanoparticles, thiazoles, heterogeneous catalysis

## Abstract

Calcium oxide (CaO) nanoparticles have recently gained much interest in recent research due to their remarkable catalytic activity in various chemical transformations. In this article, a chitosan calcium oxide nanocomposite was created by the solution casting method under microwave irradiation. The microwave power and heating time were adjusted to 400 watts for 3 min. As it suppresses particle aggregation, the chitosan (CS) biopolymer acted as a metal oxide stabilizer. In this study, we aimed to synthesize, characterize, and investigate the catalytic potency of chitosan–calcium oxide hybrid nanocomposites in several organic transformations. The produced CS–CaO nanocomposite was analyzed by applying different analytical techniques, including Fourier-transform infrared spectroscopy (FTIR), X-ray diffraction (XRD), and field-emission scanning electron microscopy (FESEM). In addition, the calcium content of the nanocomposite film was measured using energy-dispersive X-ray spectroscopy (EDS). Fortunately, the CS–CaO nanocomposite (15 wt%) was demonstrated to be a good heterogeneous base promoter for high-yield thiazole production. Various reaction factors were studied to maximize the conditions of the catalytic technique. High reaction yields, fast reaction times, and mild reaction conditions are all advantages of the used protocol, as is the reusability of the catalyst; it was reused multiple times without a significant loss of potency.

## 1. Introduction

Organic synthesis is one of the most important areas where catalysts are used, especially on a commercial scale, because they allow for a more cost-effective preparation of various organic compounds than would otherwise be possible. In their natural state, organo-catalysts are homogenous, but when modified as nanocatalysts, they exhibit heterogeneous catalytic behavior [1]. Although popular traditional catalysts such as pyridine and piperidine as well as triethyl amine have been utilized in organic synthesis transformations for many years, nanoparticles are currently being used for greener and novel organic reactions [2,3,4,5,6].

Nanocatalysts have recently piqued the interest of many academics as they have emerged as a viable, long-term, and low-cost alternative to the old commercial catalysts widely utilized in organic synthesis [4]. They display increased catalytic activity and regioselectivity in several chemical transformations due to their particle size (1–100 nm) and, therefore, their highly exposed surface area with a larger number of reactive sites, which causes the reaction to run to completion with a higher efficiency. Another advantage is the insolubility of nanocatalysts in reaction mixtures, which makes the separation, recovery, and reuse of the nanocatalysts easier. For the above reasons, many researchers have attempted to create new hybrid materials containing metal oxide nanoparticles due to their promising catalytic capabilities in various organic reactions [7,8].

Although calcium oxide (CaO) is a plentiful, environmentally friendly, and non-corrosive substance, its utility remains limited in organic reactions. Due to its low solubility, high alkalinity, and—most importantly—its ability to be recycled following the reaction, it is simple to handle. As a result, it has been identified as one of the heterogeneous catalysts most frequently utilized in the manufacture of biodiesel [9]. In addition, calcium oxide nanoparticles have been efficiently used as base catalysts in many organic transformations such as the green synthesis of polyfunctionalized pyridine derivatives [10], dihydropyrazole derivatives [11], chalcones [12], and 4H-pyran derivatives [13] as well as in the hydrogenation of ethylene [14].

Chitosan and its derivatives can be used as efficient, recyclable, and environmentally friendly catalyst supports in various applications [15,16,17,18,19,20]. Khaled et al. [18] investigated the catalytic activity of chitosan-grafted poly(2-cyano-1-(pyridine-3yl)allyl acrylate)-Cs-g-PCPA with 64% grafting as a novel, efficient, eco-friendly, and recyclable biocatalyst for Michael addition reactions and examined the use of the iron complex of the grafted chitosan as a promoter for alkyl oxidation reactions.

Due to its strong adsorption capacity through the NH_2_ and OH binding sites, chitosan, a natural polysaccharide, has been widely exploited as an efficient stabilizer for immobilizing metal oxide nanoparticles in recent decades [21,22,23,24]. A chitosan–CuO nanocomposite was prepared via the simple solution cast method and successfully used as a heterogenous basic catalyst in the synthesis of 1,3,4-pyrazoles [25] and 1,2,3-triazoles [26]. Moreover, chitosan–MgO nanocomposites were efficiently used as a heterogeneous basic catalyst in Michael additions [27] and also for the synthesis of thiazoles and 1,3,4-thiadiazoles in a comparative study with a traditional catalyst [28]. 

Abdel-Naby and coworkers synthesized a chitosan–Al_2_O_3_ nanocomposite under microwave conditions; the nanocomposite was then used as a green heterogeneous catalyst to synthesize novel imidazopyrazolyl thione derivatives [29]. Recently, Khalil et al. [30] investigated the catalytic activity of a chitosan–SrO nanocomposite as a powerful base promoter for the synthesis of 2-hydrazono-1,3,4-thiadiazole derivatives.

Shorter reaction times, a simple experimental approach, excellent yields, greater selectivity, and clean processes are all advantages of ultrasound-assisted reactions [31,32,33]. One of the many advantages of ultrasonic irradiation is its ability to play a critical role in chemistry, especially in cases where conventional methods demand extreme conditions or extended reaction times [34,35,36].

The 1,3-thiazole core structure, on the other hand, has been widely investigated in identifying new lead compounds for drug discovery. Thiazole-containing therapies such as tiazofurin (an inhibitor of IMP dehydrogenase) [37], dasatinib (a Bcr-Abl tyrosine kinase inhibitor) [38], dabrafenib (an inhibitor of enzyme B-RAF) [39], ixabepilone (microtubule stabilization) [40], and epothilone (microtubule function inhibition) (Figure 1) are among the best-selling drugs or have served as lead structures [41]. Several drug discovery teams have investigated the possible applicability of a thiazole scaffold in the design of anti-cancer drugs [42,43,44,45].

Considering the results above, and in continuation with our studies on nanocatalysis [7,8,25,26,27,28,29,30], herein, we introduce an eco-friendly protocol for synthesizing 1,3-thiazole derivatives in the presence of chitosan–CaO nanocomposites as an efficient and recyclable base catalyst. 

## 2. Materials and Methods

### 2.1. Apparatus and Instrumentation

Details were inserted in the Appendix A.

### 2.2. Preparation of the Chitosan–Based CaO Nanocomposite Films

A chitosan solution (2%, *w*/*v*) was prepared by swirling chitosan (CS) in a 1% (*v*/*v*) acetic acid solution on a magnetic stirrer for 12 h at room temperature. After completely dissolving, the pH of the resulting CS solution was adjusted to the range of 6–7 by adding a calculated amount of 1 M NaOH solution under continuous stirring. Portion-by-portion, a suspension of the estimated amount of the CaO nanopowder in a tiny amount of double-distilled water was then added to the CS solution under continuous stirring. After being cast into a 100 mm Petri dish and dried overnight at 70 °C to remove any traces of acetic acid, the mixture was microwaved for 3 min at 400 watts. The CS–CaO nanocomposite film was then removed. After completely drying, the CS–CaO nanocomposite film was removed, rinsed with distilled water, and dried at 70 °C. The characterizations were performed using the formed films.

### 2.3. Synthesis of Thiazole Derivatives ***5a,b***

In 30 mL of EtOH, a mixture of 2-(4-formyl-3-methoxyphenoxy)-N-phenylacetamide derivatives **3a,b** (10 mmol), and thiosemicarbazide **4** (0.91 g, 10 mmol) was treated with catalytic quantities of concentrated hydrochloric acid. An ultrasonic generator was used to irradiate the reaction mixture in a water bath at 50 °C for 20 min. After cooling, the precipitate was filtered, washed with ethanol, and recrystallized from acetic acid to produce thiosemicarbazones **5a,b**. 


**2-(4-((2-Carbamothioylhydrazineylidene)methyl)-3-methoxyphenoxy)-N-(4-chlorophenyl)acetamide (5a).**


A yellowish-white solid of 74% yield; m.p. 214–216 °C; IR (KBr): *v* 3403, 3350, 3312, 3277 (2NH, NH_2_), 3042, 2926 (CH), 1682 (C=O), 1607 (C=N) cm^−1^; ^1^H NMR (DMSO-*d*_6_): *δ* 3.89 (s, 3H, OCH_3_), 4.73 (s, 2H, –CH_2_O), 6.91–7.89 (m, 9H, Ar–H, NH_2_), 8.51 (s, 1H, CH=N), 10.35 (br s, 1H, NH), 11.49 (br s, 1H, NH) ppm; MS *m*/*z* (%): 392 (M^+^, 26). Elemental analysis calculated for C_17_H_17_ClN_4_O_3_S (392.86): C, 51.97; H, 4.36; N, 14.26. Found: C, 51.81; H, 4.27; N, 14.14%.


**N-(4-Bromophenyl)-2-(4-((2-carbamothioylhydrazinylidene)methyl)-3-methoxyphenoxy)acetamide (5b)**


A yellowish-white solid of 77% yield; m.p. 201–203 °C; IR (KBr): *v* 3411, 3369, 3327, 3295 (2NH, NH_2_), 3051, 2923 (CH), 1686 (C=O), 1613 (C=N) cm^−1^; ^1^H NMR (DMSO-*d*_6_): *δ* 3.87 (s, 3H, OCH_3_), 4.70 (s, 2H, –CH_2_O), 6.95–7.80 (m, 9H, Ar–H, NH_2_), 8.47 (s, 1H, CH=N), 10.29 (br s, 1H, NH), 11.53 (br s, 1H, NH) ppm; MS *m*/*z* (%): 437 (M^+^, 54). Elemental analysis calculated for C_17_H_17_BrN_4_O_3_S (437.31): C, 46.69; H, 3.92; N, 12.81. Found: C, 46.58; H, 3.82; N, 12.69%.

### 2.4. Synthesis of Thiazole Derivatives ***8a–g***


Method A: A few drops of triethylamine TEA were added to a mixture of equimolar volumes of suitable hydrazonoyl chlorides **6a–d** (1 mmol) and thiosemicarbazones **5a,b** (1 mmol) in 20 mL EtOH. An ultrasonic generator was used to irradiate the produced solution in a water bath at 50 °C for 20–60 min (irradiation was continued until all of the starting materials disappeared and the product was formed; this was monitored by TLC). After cooling, the red precipitate was filtered off, washed with EtOH, dried, and recrystallized from EtOH to yield the thiazoles **8a–g**. Below are the physical constants for the products **8a–g**.

Method B: A CS–CaO nanocomposite film (0.26 g, 15 wt%) was added to a mixture of equimolar proportions of suitable hydrazonoyl chlorides **6a–d** (1 mmol) and thiosemicarbazones **5a,b** (1 mmol) in 20 mL EtOH. An ultrasonic generator was used to irradiate the produced solution in a water bath at 50 °C for 20–40 min. To remove the CS–CaO, the heated solution was filtered and the surplus solvent was extracted under a reduced pressure. The residue was treated with methanol to obtain authentic samples of compounds **8a–g** and the formed solid was filtered and recrystallized from EtOH (m.p., mixed m.p., IR, and TLC).


**N-(4-Chlorophenyl)-2-(3-methoxy-4-((2-(4-methyl-5-((E)-phenyldiazenyl)thiazol-2-yl)hydrazono)methyl)phenoxy)acetamide (8a).**


A red solid; m.p. 151–153 °C; IR (KBr): *v* 3406, 3327 (2NH), 3043, 2929 (CH), 1683 (C=O), 1611 (C=N) cm^−1^; ^1^H NMR (DMSO-*d*_6_): *δ* 2.55 (s, 3H, CH_3_), 3.85 (s, 3H, OCH_3_), 4.71 (s, 2H,–CH_2_O), 6.91–8.2 (m, 12H, Ar–H), 8.56 (s, 1H, CH=N), 10.37 (br s, 1H, NH), 11.34 (br s, 1H, NH) ppm; MS *m*/*z* (%): 535 (M^+^, 71). Elemental analysis calculated for C_26_H_23_ClN_6_O_3_S (535.02): C, 58.37; H, 4.33; N, 15.71. Found: C, 58.29; H, 4.20; N, 15.58%. 


**N-(4-Chlorophenyl)-2-(3-methoxy-4-((2-(5-((4-methoxyphenyl)diazenyl)-4-methylthiazol-2-yl)hydrazono)methyl)phenoxy)acetamide (8b).**


A red solid; m.p. 137–139 °C; IR (KBr): *v* 3412, 3315 (2NH), 3051, 2924 (CH), 1677 (C=O), 1606 (C=N) cm^−1^; ^1^H NMR (DMSO-*d*_6_): *δ* 2.53 (s, 3H, CH_3_), 3.78 (s, 3H, OCH_3_), 3.83 (s, 3H, OCH_3_), 4.75 (s, 2H, –CH_2_O), 6.90–8.16 (m, 11H, Ar–H), 8.54 (s, 1H, CH=N), 10.58 (br s, 1H, NH), 11.33 (br s, 1H, NH) ppm; MS *m*/*z* (%): 565 (M^+^, 52). Elemental analysis calculated for C_27_H_25_ClN_6_O_4_S (565.04): C, 57.39; H, 4.46; N, 14.87. Found: C, 57.27; H, 4.41; N, 14.68%.


**N-(4-Chlorophenyl)-2-(4-((2-(5-((4-chlorophenyl)diazenyl)-4-methylthiazol-2-yl)hydrazono)methyl)-3-methoxyphenoxy)acetamide (8c).**


A red solid; m.p. 170–172 °C; IR (KBr): *v* 3403, 3294 (2NH), 3071, 2931 (CH), 1679 (C=O), 1609 (C=N) cm^−1^; ^1^H NMR (DMSO-*d*_6_): *δ* 2.54 (s, 3H, CH_3_), 3.83 (s, 3H, OCH_3_), 4.71 (s, 2H, –CH_2_O), 6.86–8.18 (m, 11H, Ar–H), 8.56 (s, 1H, CH=N), 10.27 (br s, 1H, NH), 11.34 (br s, 1H, NH) ppm; MS *m*/*z* (%): 569 (M^+^, 28). Elemental analysis calculated for C_26_H_22_Cl_2_N_6_O_3_S (569.46): C, 54.84; H, 3.89; N, 14.76. Found: C, 54.89; H, 3.77; N, 14.59%.


**N-(4-Chlorophenyl)-2-(4-((2-(5-((2,4-dichlorophenyl)diazenyl)-4-methylthiazol-2-yl)hydrazono)methyl)-3-methoxyphenoxy)acetamide (8d).**


A red solid; m.p. 177–179 °C; IR (KBr): *v* 3408, 3329 (2NH), 3020, 2933 (CH), 1684 (C=O), 1615 (C=N) cm^−1^; ^1^H NMR (DMSO-*d*_6_): *δ* 2.56 (s, 3H, CH_3_), 3.83 (s, 3H, OCH_3_), 4.76 (s, 2H, –CH_2_O), 6.88–8.03 (m, 10H, Ar–H), 8.21 (s, 1H, CH=N), 10.32 (br s, 1H, NH), 11.33 (br s, 1H, NH) ppm; MS *m*/*z* (%): 603 (M^+^, 57). Elemental analysis calculated for C_26_H_21_Cl_3_N_6_O_3_S (603.91): C, 51.71; H, 3.50; N, 13.92. Found: C, 51.64; H, 3.43; N, 13.81%.


**N-(4-Bromophenyl)-2-(3-methoxy-4-((2-(4-methyl-5-(phenyldiazenyl)thiazol-2-yl)hydrazono)methyl)phenoxy)acetamide (8e).**


A red solid; m.p. 145–147 °C; IR (KBr): *v* 3434, 3285 (2NH), 3045, 2937 (CH), 1675 (C=O), 1608 (C=N) cm^−1^; ^1^H NMR (DMSO-*d*_6_): *δ* 2.55 (s, 3H, CH_3_), 3.83 (s, 3H, OCH_3_), 4.80 (s, 2H, –CH_2_O), 6.90–8.02 (m, 12H, Ar–H), 8.56 (s, 1H, CH=N), 10.36 (br s, 1H, NH), 11.33 (br s, 1H, NH); MS *m*/*z* (%): 579 (M^+^, 52). Elemental analysis calculated for C_26_H_23_BrN_6_O_3_S (579.47): C, 53.89; H, 4.00; N, 14.50. Found: C, 53.73; H, 4.13; N, 14.37%.


**N-(4-Bromophenyl)-2-(3-methoxy-4-((2-(5-((4-methoxyphenyl)diazenyl)-4-methylthiazol-2-yl)hydrazono)methyl)phenoxy)acetamide (8f).**


A red solid; m.p. 155–157 °C; IR (KBr): *v* 3406, 3290 (2NH), 3041, 2934 (CH), 1679 (C=O), 1613 (C=N) cm^−1^; ^1^H NMR (DMSO-*d*_6_): *δ* 2.52 (s, 3H, CH_3_), 3.77 (s, 3H, OCH_3_), 3.82 (s, 3H, OCH_3_), 4.84 (s, 2H, –CH_2_O), 6.89–8.02 (m, 11H, Ar–H), 8.17 (s, 1H, CH=N), 10.64 (br s, 1H, NH), 11.34 (br s, 1H, NH) ppm; MS *m*/*z* (%): 609 (M^+^, 37). Elemental analysis calculated for C_27_H_25_BrN_6_O_4_S (609.49): C, 53.21; H, 4.13; N, 13.79. Found: C, 53.03; H, 4.07; N, 13.64%.


**N-(4-Bromophenyl)-2-(4-((2-(5-((4-chlorophenyl)diazenyl)-4-methylthiazol-2-yl)hydrazono)methyl)-3-methoxyphenoxy)acetamide (8g).**


A red solid; m.p. 181–183 °C; IR (KBr): *v* 3436, 3293 (2NH), 3049, 2928 (CH), 1682 (C=O), 1609 (C=N) cm^−1^; ^1^H NMR (DMSO-*d*_6_): *δ* 2.55 (s, 3H, CH_3_), 3.83 (s, 3H, OCH_3_), 4.78 (s, 2H, –CH_2_O), 6.88–8.03 (m, 11H, Ar–H), 8.19 (s, 1H, CH=N), 10.25 (br s, 1H, NH), 11.34 (br s, 1H, NH) ppm; MS *m*/*z* (%): 613 (M^+^, 55). Elemental analysis calculated for C_26_H_22_BrClN_6_O_3_S (613.91): C, 50.87; H, 3.61; N, 13.69. Found: C, 50.74; H, 3.51; 5.77; N, 13.50%.

## 3. Results and Discussion

### 3.1. Preparation of the Chitosan–CaO Nanocomposite 

The CS–CaO nanocomposite was prepared using a simple solution casting process [25,26,28] under microwave irradiation, as indicated in Figure 1.

#### 3.1.1. FTIR Characterization

As shown in Figure 2, the FTIR spectra for chitosan, the CaO nanoparticles, and the chitosan–CaO nanocomposites (C) were measured. The spectrum of pure CS [25,26,27,28] revealed a broad band at 3426 cm^−1^ due to the intermolecular H-bonding of the O–H and NH_2_ stretching bands positioned in the same region. The amide characteristic bands could be seen at υ = 1655 and 1606 cm^−1^ whilst the CH bands along the CS chain appeared clearly at 2915, 2876, and 1372 cm^−1^ in the spectrum. As shown in Figure 2B, the CaO nanoparticles showed two strong bands at υ = 605 and 516 cm^−1^ corresponding with the expected Ca–O stretching vibrations in the range of 620–420 cm^−1^ as reported in [46], which has been attributed to the monoclinic phase of CaO nanoparticles. The FTIR spectra of the CaO sample showed additional bands that corresponded with the CaCO_3_ source; these peaks appeared in the range of 2000 and 1200 cm^−1^, which are the characteristic peaks of the C–O stretching and bending modes of CaCO_3_ [47]. The broad band at >3300 cm^−1^ could be attributed to the OH stretching vibration from water. Although the CaCO_3_ raw material was calcined to high temperatures greater than 800 °C for 10 h, those bands still appeared. Figure 2C depicts the noisy-like shape of the hybrid chitosan–CaO nanocomposite. The presence of the combination of chitosan and CaO characteristic bands at 3400 cm^−1^, two distinct bands at 2918 and 2875 cm^−1^, and the visible alteration in the chitosan fingerprint region (especially by the CaO bands) at 629 and 538 cm^−1^ was strong evidence of structural changes due to the incorporation of the calcium oxide nanoparticles.

#### 3.1.2. X-ray Diffraction (XRD)

Figure 3 shows the structural characteristics of the native chitosan and the chitosan–CaO nanocomposite from the application of the XRD technique. Chitosan displayed two typical peaks, one strong at 2θ = 19–21° and the other weak at 2θ = 36°, which conformed with the values in the literature of the hydrated crystalline structure of chitosan [8,25,26,27,28]. The same distinctive peaks of both chitosan (2θ = 19–21°) and the CaO nanoparticles (2θ = 34° and 38.5°) [47] were disturbed with a clear shift in pattern, indicating the interaction of the CaO molecules with the chitosan chain. The Debye–Scherrer formula [48] was used to calculate the average grain size from the XRD patterns:(1)$$D\left(\mathrm{nm}\right)=\frac{-0.9\lambda }{\beta \;\cos\theta }$$
where *D* (nm) represents the crystalline size in nm and λ is the wavelength of Cu-kα_1_ = 1.54060 Å. For the CS–CaO nanocomposite pattern, β could be determined for the most intense peak. The average particle size was calculated to be 42.2 nm using this equation.

#### 3.1.3. FESEM and Morphological Changes

A SEM instrument was used to investigate the morphological alterations of the CS–CaO nanocomposites compared with unmodified chitosan. As illustrated in Figure 4, the non-porous, smooth membranous phase of chitosan consisted of dome-shaped orifices, microfibrils, and crystallites [8,25,26,27,28]. However, due to the coordination of the chitosan binding sites with the CaO molecules, the picture of the CS–CaO nanocomposite revealed a drastic morphological change.

#### 3.1.4. Energy-Dispersive X-ray Spectroscopy (EDS) and Estimation of Calcium

The presence of Ca within the chitosan matrix was confirmed by the EDS graph of the chitosan–CaO nanocomposites (Figure 5), which revealed characteristic Ca signals. The calcium content in the manufactured sample was found to be 15.54 wt%.

### 3.2. Synthesis of Thiazole Derivatives Using the CS–CaO Nanocomposite Film as Basic Heterogeneous Catalyst 

In continuation of our previous work, which has designed and synthesized bioactive heterocyclic compounds under mild conditions [26,33,35,49,50,51,52,53,54], here, we wished to report a mild and efficient procedure for synthesizing novel thiazoles. Initially, the active key thiosemicarbazone derivatives **5a,b** were prepared by the condensation of 2-(4-formyl-3-methoxyphenoxy)-N-phenylacetamide derivatives **3a,b** [55] with thiosemicarbazide **4** in an acidic ethanol solution (Figure 2). The structure of thiosemicarbazone derivatives **5a,b** were assured, depending on the data extracted from the spectra (IR, ^1^H NMR, and MS). 

New thiazole derivatives **8a–g** were synthesized from the reaction of an equimolar quantity of thiosemicarbazone derivatives **3a,b** with an equivalent of 2-oxo-*N*-arylpropanehydrazonoyl chloride **4a–g** in EtOH in the presence of two different bases (CS or CS–CaO) (Figure 1). The chemical structure of all newly produced thiazoles **8a–g** was confirmed based on spectral and elemental investigations. Their structure was confirmed by ^1^H NMR of all isolated **8a–g** derivatives, which displayed the predicted signals for the postulated structure. For instance, we detected the characteristic four singlet signals in the ^1^H NMR of compound **8a** at 2.55 (CH_3_), 3.85 (OCH_3_), 2.55 (CH_3_), 4.71 (–CH_2_O), and 8.56 (CH=N) in addition to exchangeable protons at 10.37 and 11.34 due to two NHs and twelve multiplet aromatic protons at 6.91–8.20 ppm. The ^13^C NMR of the same derivative **8a** revealed characteristic 22 non-equivalent carbon signals, as illustrated in the experimental section.

Thin-layer chromatography (TLC) was used to track the progress of all the reactions. The study for the best basic catalyst began at the outset (Table 1).

As shown in Table 1, the CS–CaO nanocomposite was a better basic catalyst than traditional TEA under ultrasonic irradiation. When triethylamine was replaced with the CS–CaO nanocomposite, the yields of the desired products of **8a–g** increased; the reaction time decreased under the same reaction conditions.

We discovered the optimal experimental conditions and variables (such as the catalyst loading, temperature, solvent, and reaction duration) for the reaction of 5 + 6 in the presence of a catalytic quantity of the CS–CaO nanocomposite under USI to obtain thiazole derivative **8a**.

The effect of the amount of catalyst on the synthesis of component 8a was investigated in the first step (Table 2, entries 1–3). The best results (91%) were obtained with a catalyst concentration of 20 mol% (Table 2, entry 3). Lower yields were achieved by using less catalyst (Table 2). The efficiency of the various solvents was then investigated using USI (Table 2, entries 3, 4, and 5). The production of product **8a** proceeded with the best yield with the fastest reaction rate in EtOH, according to a screening of several solvents (Table 2, entry 3).

In addition, the reaction time was evaluated under USI (Table 2, entries 3, 6, and 7). The best time to form product **8a** was 23 min (Table 2, entry 3).

Moreover, the influence of temperature on the reaction was investigated; the results are provided in Table 2 (entries 3, 8, 9, and 10). Table 2 shows that by increasing the reaction temperature from 25 to 35 to 50 °C whilst using USI increased the product yields from 73 to 87 to 91%, respectively. Finally, 40 °C was chosen as the ideal temperature (Table 2, entry 3).

Moreover, the recyclability of the CS–CaO nanocomposite as a basic catalyst was also examined. After each run, the catalyst film was rinsed with distilled water and dried at 60 °C for 30 min before reuse. Under ideal conditions, the catalyst was reused three times without a significant loss of catalytic performance (15 wt% and 23 min; Table 3).

In other words, the catalyst retained almost 93.4% of its performance after three runs, which indicated less than a 1% reduction after each run. In the fourth run, it lost about 50% of its performance.

The optimum reaction conditions for the synthesis of product **8a**, as indicated in Table 2, were a reaction of **5a + 6a** in EtOH under USI in the presence of the CS–CaO nanocomposite (15 wt%) at 40 °C for 23 min. Thus, the irradiation of **5a,b** + **6b–d** under the optimum conditions led to the formation of thiazole derivatives **8b–g** (Figure 1).

## 4. Conclusions

This study used the microwave-assisted solution casting method to efficiently prepare a chitosan–CaO nanocomposite. The produced nanocomposite film was thoroughly examined using FTIR, XRD, FESEM, and EDS measurements. All the data confirmed the presence of calcium oxide nanoparticles within the chitosan matrix. The FTIR spectra showed an obvious change, especially in the fingerprint region attributed to Ca–O bending vibrations. A combination of chitosan and CaO characteristic peaks could be seen in the XRD pattern. The SEM image of the nanocomposite also revealed a clear uniform surface alteration of the chitosan upon coordination with the CaO molecules. The CS–CaO hybrid nanocomposite film was efficiently used as an eco-friendly heterogeneous basic catalyst in synthesizing thiazole derivatives, which has a significant industrial impact. Thus, the invented catalyst is promising due to its non-toxic nature and economic impact and it may be used in the industrial production of the reported compounds. We concluded that the nanocomposite basic catalyst could be used to efficiently synthesize a variety of heterocycles that have previously been manufactured via non-green methods.

## Data Availability

Data are contained within the article.

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
