# Peer review of "Efficient, Recyclable, and Heterogeneous Base Nanocatalyst for Thiazoles with a Chitosan-Capped Calcium Oxide Nanocomposite"

_polymers, 2022, doi:10.3390/polym14163347_

Round 1

Reviewer 1 Report

The manuscript by Khalil et al. reports the use of CaO nanoparticles capped by chitosan as base catalyst for the production of thiazole. The use of such nanocomposite and the results are interesting, however the presentation and discussion of the results should be definitely improved and some lacking information should be provided.

1.      Are there previous studies using CaO as heterogeneous catalyst for similar reactions? In this case they should be mentioned at least in the Introduction.

2.      Synthesis: what is the amount of CaO used? Does the nominal content of CaO correspond to the measured value?

3.      FTIR characterization: since the two bands at 2118 and 2359 cm-1 in the composite are considered as evidence for the coordination, at least a hypothesis of the attribution of these vibrations should be given. In other words, what is the expected interaction between the chitosan functional groups and CaO nanoparticles surface? Moreover, what is the origin of the additional bands seen in the spectrum of bare CaO nanoparticles?

4.      The nanocomposites are prepared in the form of films. Are they used as such or as powders in the catalytic runs? Which is the catalyst loading? It is expressed as mol%, but for the heterogeneous catalyst it is not clear how the moles are calculated.

5.      There is no reference catalytic experiment with bare CaO without chitosan to justify the role of the polymer in improving the efficiency and/or stability of the catalyst

6.      The quality of Figs. 2, 3 and 5 is very low.

7.      Some NMR spectra of the products should be probably reported as supporting material.

Author Response

Responses to the reviewer comments

Authors would like to thank the reviewer for his valuable comments.

Please see the following table of our responses, corrections and explanation based on the reviewer comments.

Reviewer 1 Comments

Our Reply

1

Are there previous studies using CaO as heterogeneous catalyst for similar reactions? In this case they should be mentioned at least in the Introduction.

Although, calcium oxide (CaO) is a plentiful, environmentally friendly, and non-corrosive substance, but its utility is still very limited in few of organic reactions. Only it is frequently used in biodiesel production, so statement referring to this information is added to the introduction part. Some citations are added to the introduction part.

2

Synthesis: what is the amount of CaO used? Does the nominal content of CaO correspond to the measured value?

The used catalyst in this study contains 15% wt./ chitosan and the amount was estimated by EDS analysis.

3

FTIR characterization: since the two bands at 2118 and 2359 cm-1 in the composite are considered as evidence for the coordination, at least a hypothesis of the attribution of these vibrations should be given. In other words, what is the expected interaction between the chitosan functional groups and CaO nanoparticles surface? Moreover, what is the origin of the additional bands seen in the spectrum of bare CaO nanoparticles?

Thank you very much for your comment, Explanation to your comments about the findings in FTIR is added to the FTIR characterization.

4

The nanocomposites are prepared in the form of films. Are they used as such or as powders in the catalytic runs? Which is the catalyst loading? It is expressed as mol%, but for the heterogeneous catalyst it is not clear how the moles are calculated.

The catalyst is shaped in the form of films and strips of this film is added to the reaction medium during the reaction, after reaction completion this film is simply removed and washed with water and alcohol for re-use.

Thus, amount of catalyst (1 g) is clarified in the experimental section.

5

There is no reference catalytic experiment with bare CaO without chitosan to justify the role of the polymer in improving the efficiency and/or stability of the catalyst

Actually, a model reaction was conducted with bare CaO but it was found that it so difficult to remove the catalyst and purify the reaction product. Thus, the Yield calculation were unacceptable and inaccurate. 

6

The quality of Figs. 2, 3 and 5 is very low.

The figures are replaced with ones of better quality.

7

Some NMR spectra of the products should be probably reported as supporting material.

Done (see supporting information file)

Reviewer 2 Report

-The introduction is poor. You barely mention anything about chitosan-catalyst hybrid material preparation and application. Chitosan is widely used for the preparation of hybrid material by many researchers, and you exclusively autocite 3 articles.

Lines 54-55 “Because of its strong adsorption capacity through the NH2 and OH binding sites, chitosan, a natural polysaccharide, has been widely exploited as an efficient stabilizer for immobilizing metal oxide nanoparticles in recent decades [9-11]

This part of the introduction should be widely expanded, and you should cite proper bibliography.

2.4. Synthesis of thiazole derivatives 8a-g.

“A few drops of TEA” What is TEA?  are a few drops a quantitative quantity of catalyst?

                -Figure 2 has a bad quality in several aspects.

Numbers in spectra B and C are deformed and they have bad quality compared to spectra A.

Peak positions have to be revised. Peak at 605 cm-1 (spectra b) is left-positioned compared to peak at 629 cm-1 (spectra c). This can be possible.

Y-axis scale (transmittance %) should be included.

Line 194-199 “with the presence of a broad band at 3400 cm-1 and two distinct bands at 2118 and 2359 cm-1, which are strong evidence for CaO molecules coordinating within the binding sites (NH2 and OH groups) along the chitosan backbone. Finally, the visible alteration in the chitosan fingerprint region, especially at 629 and 538 cm-1, is strong evidence of structural changes due to the incorporation of calcium oxide nanoparticles. 199” Any citation for this.

                -XRD

Line 203-204 “Figure 3 shows the structural characteristics of native chitosan (A), pure CaO nanoparticles (B), and chitosan-CaO nanocomposite (C) applying the XRD technique” Spectra C is missing in figure 3

                -EDS Ca content. Line 230 “calcium content in the manufactured sample was found to be 15.54% wt”. This value of calcium content doesn’t match with the value in figure 5.

                -Table 1: Why is the reaction time so variable? Why not a fixed time?

-Did you perform the catalytic reaction with CaO the nanoparticles (unsupported)?

-Figure of re-used catalysis should be removed. Table 3 gives exactly the same information. Moreover, table 3 is clear.

-Characterization of chitosan-CuO is limited. Additional characterization could be good for a better understanding of the catalyst.

Author Response

Responses to the reviewer comments

Authors would like to thank the reviewer for his valuable comments.

Please see the following table of our responses, corrections and explanation based on the reviewer comments.

Reviewer 2 Comments

Our Reply

1

The introduction is poor. You barely mention anything about chitosan-catalyst hybrid material preparation and application. Chitosan is widely used for the preparation of hybrid material by many researchers, and you exclusively autocite 3 articles. Lines 54-55 “Because of its strong adsorption capacity through the NH2 and OH binding sites, chitosan, a natural polysaccharide, has been widely exploited as an efficient stabilizer for immobilizing metal oxide nanoparticles in recent decades [9-11]”

Although, calcium oxide (CaO) is a plentiful, environmentally friendly, and non-corrosive substance, but its utility is still very limited in organic syntheses. Only it is frequently used in biodiesel production, so statement referring to this information is added to the introduction. Ref. 9 

Also, more references are added to show the catalytic potency in organic syntheses.

2

2.4. Synthesis of thiazole derivatives 8a-g.

“A few drops of TEA” What is TEA?  are a few drops a quantitative quantity of catalyst?

TEA is triethylamine 

We used quantitative amount of TEA (1 mmol, 0.07 mL)

Ok, the abbreviation is clarified in the experimental part.

3

-Figure 2 has a bad quality in several aspects.

Numbers in spectra B and C are deformed and they have bad quality compared to spectra A.

Peak positions have to be revised. Peak at 605 cm-1 (spectra b) is left-positioned compared to peak at 629 cm-1 (spectra c). This can be possible.

Y-axis scale (transmittance %) should be included.

The figure is corrected and its quality is enhanced.

4

Line 194-199 “with the presence of a broad band at 3400 cm-1 and two distinct bands at 2118 and 2359 cm-1, which are strong evidence for CaO molecules coordinating within the binding sites (NH2 and OH groups) along the chitosan backbone. Finally, the visible alteration in the chitosan fingerprint region, especially at 629 and 538 cm-1, is strong evidence of structural changes due to the incorporation of calcium oxide nanoparticles. 199” Any citation for this.

Thank you for your valuable notice. Actually, the spectra were repeated twice and the additional peaks at 2118 and 2359 appeared but in very lower intensity, thus we cancelled out this argument in the discussion of FTIR. Thus, based on the combination of chitosan and CaO characteristic bands, we consider the presence of two characteristic peaks of CaO in the finger print region with slight shift as evidence for the interaction between chitosan and CaO.

5

-XRD

Line 203-204 “Figure 3 shows the structural characteristics of native chitosan (A), pure CaO nanoparticles (B), and chitosan-CaO nanocomposite (C) applying the XRD technique” Spectra C is missing in figure 3

Thanks, the figure caption is corrected according to the available data. In XRD, we studied the change in chitosan crystallinity upon mixing with CaO.  

6

-EDS Ca content. Line 230 “calcium content in the manufactured sample was found to be 15.54% wt”. This value of calcium content doesn’t match with the value in figure 5.

The error is corrected, the obtained amount of Ca was 15.54 as in the inserted table, in Figure 5

7

-Table 1: Why is the reaction time so variable? Why not a fixed time?

Variable reaction times because each reaction need its actual time to attain complete reaction as shown using TLC.  Fixed time may be giving some incomplete reactions.

8

-Did you perform the catalytic reaction with CaO the nanoparticles (unsupported)?

Actually, a model reaction was conducted with bare CaO but it was found that it so difficult to remove the catalyst and purify the reaction product. Thus, the Yield calculation were unacceptable and inaccurate. 

9

-Figure of re-used catalysis should be removed. Table 3 gives exactly the same information. Moreover, table 3 is clear.

Ok, it is removed.

10

-Characterization of chitosan-CuO is limited. Additional characterization could be good for a better understanding of the catalyst.

Actually, we focused on the study of the preparation and characterization for the chitosan – CaO nanocomposite and unfortunately analytical tools are limited and take long time.

Round 2

Reviewer 1 Report

Although the authors have tried to reply to some comments, the quality of the work does not seem greatly improved and there are issues that need to be addressed. 

1. EDS spectrum (Fig. 5) has been replaced with a different spectrum which shows Copper instead of Calcium. It is hard to understand this change. 

2. The comments on FTIR spectra and on chitosan-CaO interactions are still poor and generic. 

3. Authors write "Although, calcining the CaCOraw material to high temperatures more than 800oC for 10h, still those bands appeared" (p. 6). However, it is stated in the Supplementary material that commercial CaO nanoparticles were used, so this is unclear. 

4. In Table 2 and related discussion the catalyst loading is expressed as mol% (see point 4 in the first Reviewer's report). What mol% corresponds to 1 g of catalyst? This still appears unclear. 

5. The introduction about the use of CaO as catalyst remains quite limited. 

6. In Fig. 3 the scale values are still hardly readable.  

Author Response

Responses to the reviewer comments

Authors would like to thank the reviewer for his valuable comments.

Please see the following table of our responses, corrections and explanation based on the reviewer comments.

Reviewer 1 Comments

Our Reply

1

EDS spectrum (Fig. 5) has been replaced with a different spectrum which shows Copper instead of Calcium. It is hard to understand this change. 

EDS spectrum is replaced by correct one.

2

The comments on FTIR spectra and on chitosan-CaO interactions are still poor and generic. 

FTIR is used as comparative study, and we just focused on the differences that used to clarify the formation of hybrid material and the structural changes in chitosan with incorporation of CaO molecules.

3

Authors write "Although, calcining the CaCO3 raw material to high temperatures more than 800oC for 10h, still those bands appeared" (p. 6). However, it is stated in the Supplementary material that commercial CaO nanoparticles were used, so this is unclear. 

Actually, we purchased commercial CaO as mentioned and we utilized it as such without any modification. But in order to reply the reviewer comment about the extra peaks in its IR, we explained the presence of such extra peaks as result of probable existence of some impurities of the raw material (CaCO3) that were used in CaO preparation. Just for explanation

4

In Table 2 and related discussion the catalyst loading is expressed as mol% (see point 4 in the first Reviewer's report). What mol% corresponds to 1 g of catalyst? This still appears unclear. 

We used 20 mol% of 15% chitosan-CaO nanocomposite

and 1mmol of starting materials:

moles of catalyst = 0.2 (1mmol) = 0.0002mol

M.Wt of catalyst =  15% CaO + 85% Chitosan = 0.15 (56) + 0.85 (1526.5) = 1306 g\mol

Grams of catalyst = moles x M.Wt = 0.0002 x 1306 = 0.26g

So 20mol% of catalyst  = 0.26g

Please see table 2

1g was corrected to 0.26g (for 20mol%)

5

The introduction about the use of CaO as catalyst remains quite limited. 

The introduction part is modified and uses of chitosan and CaO in reported organic syntheses are added. 

6

 In Fig. 3 the scale values are still hardly readable.  

We modified the resolution of Figure 3 scale.

Author Response

Responses to the reviewer comments

Authors would like to thank the reviewer for his valuable comments.

Please see the following table of our responses, corrections and explanation based on the reviewer comments.

Reviewer 2 Comments

Our Reply

1

The introduction is still very poor, and it doesn’t provide a sufficient background and include all relevant references. It must provide a proper background about the reaction (Thiazoles) and the preparation of nanocomposites by using Chitosan.

The introduction part is modified and uses of chitosan and CaO in reported organic syntheses are added. 

2

Lines 54-59: Use of CaO not related to the synthesis of Thiazoles - Line 61-63: I am not asking for a review about the preparation of chitosan-MOX materials, but I expect more than three lines. Do researchers use different ways to prepare this type of composites? What are the main findings of these researchers? what do they get? Reusability? Increase in catalytic activity?

Actually, CaO is used in a similar organic reaction but till now still its use is very limited, now we cited all the available reported data about its utility. In fact, this is the driving force that encourage us to investigate its utility in the synthesis of thiazoles. As we mentioned in experimental part we used the microwave assisted solution cast method. In results and discussion we showed that the nanocomposite could be separated, recovered and reused for several times.

3

Line 65-68: This is the aim/goal of your investigation. This should be at the end of the introduction

Ok, the sentence is shifted to the end of introduction

4

Line 72-79: Acceptable introduction about Thiazole - Line 88-91: This is again part of the aim/goal of your investigation.

Ok, the aim is shifted to the end of introduction part.

5

However, there is something unclear about the thiazoles that you are preparing. Are they new thiazoles? are you preparing the thiazoles mentioned in lines 72-79? I expect a complete restructuration and modification of the introduction.

All prepared thiazoles in this work are new and their structures were elucidated

Thiazoles mentioned in lines 72-79 were previously reported and cited in the introduction section

6

Table 1: Why is the reaction time so variable? Why not a fixed time? Variable reaction times because each reaction need its actual time to attain complete reaction as shown using TLC. Fixed time may be giving some incomplete reactions. This is still unclear, but I observed that you say in section 2.4 “the product was formed, monitored by TLC”. What’s TLC means? It is not possible to know the procedure for the identification and quantification of your products. Moreover, you should add the equation used for the calculation of yield (%).

TLC: Thin Layer Chromatography

TLC used to follow up the completion of reactions

Yield% = Actual product Wt \ Calculated product Wt

7

About the synthesis of thiazole “Method B: A 1 g CS-CaO nanocomposite film (15 % wt) was added to a mixture of equimolar proportions of suitable hydrazonoyl chlorides 6a-d (1 mmol) and thiosemicarbazones 5a,b (1 mmol) in 20 mL EtOH (0.1 g).“ This 0.1 g, what are you referring to that mass?

We used 20 mol% of 15% chitosan-CaO nanocomposite

and 1mmol of starting materials:

moles of catalyst = 0.2 (1mmol) = 0.0002mol

M.Wt of catalyst =  15% CaO + 85% Chitosan = 0.15 (56) + 0.85 (1526.5) = 1306 g\mol

Grams of catalyst = moles x M.Wt = 0.0002 x 1306 = 0.26g

So 20mol% of catalyst  = 0.26g

Please see table 2

0.1g was corrected to 0.26g (for 20mol%)

Round 3

Reviewer 1 Report

Following the revisions the manuscript appears suitable for publication. 

Reviewer 2 Report

The manuscript has been improved with all the corrections, especially the introduction.